# An Integrated Approach for Real-Time Monitoring of Knee Dynamics with IMUs and Multichannel EMG

**DOI:** 10.3390/s23218955

**Published:** 2023-11-03

**Authors:** Nebojsa Malesevic, Ingrid Svensson, Gunnar Hägglund, Christian Antfolk

**Affiliations:** 1Department of Biomedical Engineering, Faculty of Engineering, 223 63 Lund, Sweden; ingrid.svensson@bme.lth.se (I.S.); christian.antfolk@bme.lth.se (C.A.); 2Orthopedics, Department of Clinical Sciences, Lund University, 223 65 Lund, Sweden; gunnar.hagglund@med.lu.se; 3Department of Orthopedics, Skane University Hospital, 223 65 Lund, Sweden

**Keywords:** cerebral palsy, spasticity, knee angle measurement, inertial measurement unit, electromyography, wearable device

## Abstract

Measuring human joint dynamics is crucial for understanding how our bodies move and function, providing valuable insights into biomechanics and motor control. Cerebral palsy (CP) is a neurological disorder affecting motor control and posture, leading to diverse gait abnormalities, including altered knee angles. The accurate measurement and analysis of knee angles in individuals with CP are crucial for understanding their gait patterns, assessing treatment outcomes, and guiding interventions. This paper presents a novel multimodal approach that combines inertial measurement unit (IMU) sensors and electromyography (EMG) to measure knee angles in individuals with CP during gait and other daily activities. We discuss the performance of this integrated approach, highlighting the accuracy of IMU sensors in capturing knee joint movements when compared with an optical motion-tracking system and the complementary insights offered by EMG in assessing muscle activation patterns. Moreover, we delve into the technical aspects of the developed device. The presented results show that the angle measurement error falls within the reported values of the state-of-the-art IMU-based knee joint angle measurement devices while enabling a high-quality EMG recording over prolonged periods of time. While the device was designed and developed primarily for measuring knee activity in individuals with CP, its usability extends beyond this specific use-case scenario, making it suitable for applications that involve human joint evaluation.

## 1. Introduction

Measuring human joint kinematics and muscle activity is instrumental in comprehensively assessing movement and function. This integrated approach allows for a deeper understanding of how our bodies operate, facilitating more accurate diagnoses of musculoskeletal conditions and the development of targeted interventions for improved health and performance. Cerebral palsy (CP) is a syndrome of motor impairment that includes a large collection of movement and posture disorders [1,2,3]. CP, with a relatively stable prevalence of about two to three out of 1000 live births is the most common cause of motor disability in childhood [2,3]. CP has several etiologies and is caused by lesions in or abnormal development of the brain, including the regions that govern movement [2,4]. Over 90% of cases can be traced to the perinatal period, with risk factors including preterm birth, perinatal infection, acidosis or asphyxia, and multiple gestations, among others [4]. The clinical manifestations of CP vary and cover a wide range of abnormalities, predominantly movement disorders, but also poor balance and sensory deficits [4,5]. Spasticity (increased muscle tone), muscle weakness, and impaired postural control are some of the primary clinical presentations used in the diagnosis of CP [4].

Spasticity is one of the most common manifestations of CP and is present in 78–88% of the population with CP [6,7]. A spastic muscle will not be allowed the same amount of lengthwise excursion as its counterpart with normal tonus. Consequently, spasticity may inhibit growth in terms of the length of the muscle, resulting in the development of muscle contractures and potentially skeletal and joint deformities. These contractures, in turn, result in a decreased range of motion of the joint actuated by the muscle [8].

To prevent the development of contractures, there are several different treatment/management options [9]. An injection of botulin toxin is a widely used treatment for focal spasticity [10,11]. Other treatment options include Selective dorsal rhizotomy (SDR) and systemic antispasticity medications such as baclofen [4,8]. Management options also include physical and occupational therapy [12] and orthotic treatments that involve positioning joints in a way that counteracts the contracture [13].

Clinical measures to quantify contractures include measurements of the passive range of motion (PROM) [14]. Some form of goniometer is typically used; however, the measurement can be influenced by several factors, including the experience of the assessor, the level of cooperation from the child, and the presence of muscle spasticity [15,16]. Measurement errors of 5–15° have typically been reported for goniometric measures in the lower limb [17]. Clinical measures to quantify spasticity include the Modified Ashworth Scale, where the degree of resistance to passive movement in the relevant muscle groups is rated, and the Modified Tardieu Scale, where the range of movement recorded represents the point in the joint range where a velocity-dependent ‘catch’ was felt during a quick stretch of the relevant muscle groups [14].

In order to evaluate spasticity and contractures, objective measurements of joint angles and muscle activity are of great importance, typically defined as the range of motion (ROM) and the angle of catch [18]. Furthermore, long-term measurements are needed to determine the progression of and the efficacy of treatment of CP [19]. Although it is possible to use an optical motion-tracking system to measure joint angles, it is not a practical solution due to the constraints of marker visibility. Thus, an optical motion capture system would not be suitable for the measurements in open space (e.g., outside) and during sleeping when body parts and markers might be covered. Van den Noort et al. [18] used inertial measurement unit (IMU) sensors positioned on the thigh and the shank to measure the angle of the knee joint in a cross-sectional study. Carceff et al. [20] used IMUs positioned on the chest, thighs, and shanks to compare characteristics of gait both in the lab and daily life settings. Kim et al. [21] used a single IMU on the arm and machine-learning algorithms to determine the degree of spasticity. Van den Noort et al. [22] used IMUs positioned on the pelvis, thighs, shanks, and feet of children with spastic CP to determine joint angles while the participants were walking on a 10 m walkway at a self-selected walking speed. Bojanic et al. [23] used EMG sensors on the leg to measure the activity of muscles that extend and flex the knee and for dorsal/plantar flexion of the foot during gait in order to develop a method for tracking motor disorders in the lower limb. Xu et al. [24] used EMG sensors on muscles used in wrist extension/flexion to analyze different parameters of the EMG in children with spastic hemiplegic CP. Stackhouse et al. [25] used EMG sensors on the quadriceps femoris and triceps surae muscles to quantify muscle activation, contractile properties, and fatigability in children with CP. Michelsen et al. [26] developed a wearable textile EMG recording system to record leg muscle activity in 10 children with spastic CP.

As shown in the previous paragraph, both knee angle and muscle activity provide valuable and complementary information about the spasticity and contractures of the knee joint. However, the previous studies focused on one out of these two aspects. To fully understand and evaluate knee use and the progression of CP-induced issues, it is beneficial to simultaneously obtain the EMG signal and the resulting knee angle over prolonged periods of time, such as for 24 h. That way it is possible not only to detect insufficient stretching of the knee during a day but also to assess muscle activation patterns in dynamic and sedentary activities of daily living. Importantly, using this multimodal measurement it is possible to know if the muscles are contracted, co-contracted, or relaxed in periods when the knee was static at a certain angle, i.e., during sleeping, which was missing from the previous studies.

The aim of this work was to develop and validate an instrument capable of long-term measurements of the knee angle and of the muscle activity related to the knee joint. The goal is to have a standalone, minimal in size and weight, easy-to-don-and-doff, modular system capable of recording for 24 h outside of the lab. In this work, we report on the technical validation of the instrument on ten able-bodied participants (not on patients with CP). Although the main motivation for developing and evaluating this system is to include it in the ongoing clinical CP study, by the construction, neither the hardware nor algorithms are made specifically for patients with CP. Therefore, the presented device can be used for the measurement of any human joint activity and in any protocol.

## 2. Materials and Methods

### 2.1. Subjects

In the study aiming at evaluating the knee angle measurement device, 10 able-bodied subjects participated. The exclusion criterion was the inability of the participant to maintain a treadmill-imposed walking speed for 10 min. The study was approved by the Swedish Ethical Review Authority (DNR 2019-02452) and was conducted in accordance with the tenets of the Declaration of Helsinki. All participants were informed about the contents of the experiments, both verbally and in writing, and gave their informed and written consent.

### 2.2. Knee Angle Measurement (KAM) Device

The KAM device (see Figure 1), intended for knee activity assessments, most notably knee angle measurements, includes hardware and algorithm components, see device overview in Figure 1. The hardware was designed to fulfill the following requirements:Small footprint—using only essential electronic components and keeping the size of printed circuit boards minimal to enable integration into a garment and comfortable use over prolonged periods of time.Robust and reliable knee angle measurement—compared with mechanical angle sensors that obstruct natural ROM and/or deteriorate over time due to the mechanical strain, wear and tear, and misalignment with the axis of knee rotation, the knee angle measurement using IMU is based on two miniature integrated circuits which only have to be placed on two leg segments (thigh and shank).Muscle activity measurement—to complement the information related to the knee angle, the device integrates an EMG amplifier/digitizer capable of acquiring information related to muscle activity.Embedded microcontroller—to handle communication with IMU and EMG sensors, running knee angle calculations in real-time, and storing the data, the device has an ARM M7 microcontroller.24 h measurement sessions—as one of the goals of the study is to provide whole-day knee activity measurement, the battery of the device was chosen to enable continuous power for 24 h. As in the current hardware design, it would require a relatively large discrete battery cell, to reduce the weight of the garment, the device relies on the external battery pack of 5000 mAh that is carried in the trousers pocket.

The KAM device is made as a modular system having several boards on separate PCBs and connected via flat flexible cables (FFCs), see Figure 1.

The main board contains a Teensy 4.1 microcontroller with an ARM M7 processor, an SD card slot, and power regulators. The board was interfaced with sensor boards via 8-pin Molex Click-Mate connectors. The EMG amplifier board containing an ADS1299 chip from Texas Instruments was originally printed on the same PCB as the main board, but it has been made optionally separable from it. This way, it is possible to use them both as a single PCB, separated and connected using FFCs or without an EMG sensor if it is not necessary for a specific study or patient. The communication between the two boards was carried out using an SPI interface operating at 6 MHz. EMG signals were sampled at 1 kHz, with 24-bit amplitude resolution, and stored on the SD card in 10 min batches. Two IMU PCBs containing BNO085, Bosch, were also connected using FFCs and Molex cables. The communication was based on two SPI interface channels operating at 3 MHz. The IMUs provide quaternion values at 200 Hz. The device firmware running on the Teensy 4.1 microcontroller board was responsible for the continuous reading of sensors (IMU and EMG), calculating the knee angle, and storing the data on the SD card. The IMU quaternion signal preprocessing and angle calculation followed the methods from Malesevic et al. [27] and Siminovitch [28], where the final equation (Equation (1)) implemented in the microcontroller firmware was
(1)$$\begin{array}{c} Angle=114.59\cdot \mathrm{acos}\left(abs\left(\begin{array}{c} {quatReal}_{1}\cdot {quatReal}_{2}+{quatI}_{1}\cdot {quatI}_{2}\\ +{quatJ}_{1}\cdot {quatJ}_{2}+{quatK}_{1}\cdot {quatK}_{2} \end{array}\right)\right) \end{array}$$
where *Angle* denotes the total knee angle, *quatReal_n_* denotes the real part of the quaternion of the *n*-th sensor, *quatI_n_* denotes the *i-component* of the quaternion of the *n*-th sensor, *quatJ_n_* denotes the *j-component* of the quaternion of the *n*-th sensor, and *quatK_n_* denotes the *k-component* of the quaternion of the *n*-th sensor.

### 2.3. Evaluation

The first test of the KAM device was carried out using the industrial robot IRB120, ABB, Västerås, Sweden. This test included only the angle measurement module to estimate the angle between two IMU sensors. The IMUs were placed on two segments of the robotic arm, which was moved at a constant angular velocity within the predefined range (0–130°), see Figure 2. The three angular velocities for the robot’s degree of freedom used in this test protocol were 22°/s, 75°/s, and 150°/s. The error of range of motion (full range error) was evaluated for each angular velocity by comparing the robot range, which was set to 130°, and the range measured by KAM. In addition, to evaluate the influence of unintentional misalignment between the IMU sensors, in the repeated test one of the sensors was rotated for 10°. This evaluation was used to estimate the full range measurement error and the linearity of the angle measurement.

The KAM device was evaluated on healthy participants using an optical motion-tracking system (Qualisys, Göteborg, Sweden) that was considered the “gold standard” due to its small errors in tracking the reflective marker’s 3D position (3D resolution of 0.04 mm). The motion-tracking system comprised 12 Arqus A12 motion-tracking cameras and 2 Miqus video cameras. Arqus A12 cameras have a 12 MP optical sensor (4096 × 3072 resolution) and were collecting data at 300 frames per second. Miqus cameras were used to capture video at 25 frames per second, which were used to review the recording session in the case of unexpected results. In addition, the video was used to synchronize motion-tracking signals with the signals from KAM by detecting the turning on of the LED onboard the KAM device.

To ensure that the knee angle measured by the motion-tracking system and KAM are comparable, we made custom 3D-printed templates with slots for the IMU sensor and optical markers. For redundancy, each template had five optical markers arranged in different configurations, see Figure 3. The templates were placed on the thigh and shank and tracked as rigid bodies. Qualisys Track Manager (QTM), Qualisys, Göteborg, Sweden, was used to capture motion data and calculate relative angles between rigid bodies (templates). The QTM computed roll, pitch, and yaw angles between two templates placed on the thigh and shank. As KAM measures total knee angle, regardless of sensor placement error, the same total angle was also calculated for the motion-tracking data using the following equation (Equation (2)):(2)$$Tot\left(roll,pitch,yaw\right)=Rot\left(\hat{z},roll\right)Rot\left(\hat{y},pitch\right)Rot\left(\hat{x},yaw\right)$$
where *Tot* denotes the total knee angle and *roll*, *pitch*, and *yaw* are rotations around the *z*, *y*, and *x*-axes, respectively.

To create similar test conditions for all the subjects, the study protocol included an instrumented treadmill (Gaitway 3d 170, h/p/cosmos, Nussdorf-Traunstein, Germany) that was programmed to run a custom speed profile. During the measurement, the treadmill increased speed every 2 min, imposing three walking speeds, 2 km/h, 3 km/h, and 4 km/h, and a running speed of 7 km/h. The metrics selected for the evaluation of the KAM device included cross-correlation (Pearson cross-correlation), sample-to-sample error, and root-mean-square error (RMSE) between the knee angles measured by KAM and motion-tracking systems. As the data used to compare these two systems did not follow the normal distribution determined by the Lilliefors test, we used the Friedman test with Bonferroni post hoc correction to assess the statistically significant differences between the two signals and between metrics for different walking/running speeds.

Besides recording knee angles, a separate measurement was conducted using all KAM sensors, the 8-channel EMG, and the knee angle sensor. The aim of this measurement was to evaluate the quality of EMG signals and analyze muscle synergies and cyclical activations. Self-adhesive EMG electrodes (Red Dot 2670, 3M, Maplewood, MN, USA) were placed on the thigh (Rectus femoris, Vastus medialis, Biceps femoris, and Semitendinosus) and shank (Tibialis anterior, two heads of Gastrocnemius, and Soleus). A reference electrode was placed on the patella. EMG signals were sampled at 1 kHz while the knee angle was recorded at 100 Hz. In this protocol, the walking speed on the treadmill was self-paced and lasted for 15 min. For this analysis, the EMG signal was first bandpass filtered between 10 Hz and 500 Hz using an offline Butterworth filter of 6th order. The quality of the EMG signal was estimated as the signal-to-noise ratio of 8 EMG channels during self-paced walking using the following equation:(3)$$SNR\left(ch\right)=20*\log\left(\frac{RMS\left(signal\right)}{RMS\;\left(noise\right)}\right)$$
where *SNR* denotes the signal-to-noise ratio of an EMG channel *ch*, RMS denotes the root mean square of the EMG signal within a 250 ms window, *signal* denotes a portion of the EMG signal during a contraction (a signal window with high EMG level), and *noise* denotes a portion of the EMG signal with no apparent muscle activation (a signal window with low EMG level or rest).

## 3. Results

In the test with the industrial robot, the mean full range error (FRE) for all test conditions was 0.58°. The linearity of the sensor was evaluated using goodness-of-fit metrics: summed square of residuals (SSE), square of the correlation between the robot and KAM angle (R-square), and root-mean-square error (RMSE).

As can be seen in Table 1, the FRE was below 1° in the 130° range of motion. The value of R-squared in all of the test conditions was close to 1 (although in the figure it was rounded down to two decimals, the calculated values were closer to 1–10^−5^) indicating almost perfect linearity with the robot actuator that was used as the “ground truth”. The RMSE was also very low, confirming the high linearity of the KAM sensor.

A sample of the signals recorded using KAM and the motion-tracking system is shown in Figure 4. Due to the higher sampling rate of the motion-tracking system, the knee angle is smoother than the angle measured by KAM. In addition, although the shapes of the two angles look similar, the range of motion of the two systems is somewhat different, which is evaluated in the analysis within this paper. The statistical analysis of the signals showed that there is a statistically significant difference between the two distributions, which is expected due to the relatively large sample size (54,000 samples per subject). The meaningfulness of the difference between the signals was then evaluated using cross-correlation (Pearson), sample-to-sample error, and RMSE between knee angles measured by the KAM and motion-tracking systems.

The observed similarity of knee angle shapes was confirmed by the cross-correlation between KAM and the “gold standard“ (motion-tracking system), see Figure 5. The mean cross-correlation coefficient of all subjects was above 0.95 for each of the walking/running speeds. There was no statistically significant difference between cross-correlations at different walking/running speeds. It is interesting that the cross-correlation coefficient is highest and the variability lowest for the running speed (7 km/h), indicating a good dynamic range of IMU sensors that are able to capture the signal shape during rapid movements.

The sample-to-sample error of the measured knee angle for all subjects is shown in Figure 6. The mean sample-to-sample error of all subjects was below 4°, while in the most extreme cases, it went to 8°. A multiple comparison test did not find any statistically significant differences between the sample-to-sample error for different walking/running speeds, confirming that the accuracy of the knee angle estimation does not depend on the walking conditions.

As another evaluation metric of the KAM device, we calculated the root-mean-square error (RMSE) between KAM and the “gold standard“. The data were analyzed per walking/running speed, see Figure 7. The median RMSE of all subjects was below 8° regardless of the speed of walking/running. A multiple comparison test did not find any statistically significant differences between the RMSE for different walking/running speeds. Although the RMSE variability between different participants was lowest for running, the median RMSE was highest (though, not statistically significant).

The second test of the device focused more on the EMG signal quality. The median signal-to-noise ratio of 8 EMG channels was 108 dB (Q1—88 dB, Q3—118 dB), which is in line with state-of-the-art EMG devices ([29]). The signal-to-noise ratio was estimated using EMG signals during normal walking where the noise level was estimated during standing (just before starting to walk on the treadmill). As can be seen in Figure 8, the activity of leg muscles is clearly distinguishable and temporally synchronized with the gait phases. It is also possible to estimate muscle synergies using such synchronized kinematic and EMG recordings, which can be used to evaluate affected gait in patients with CP. In addition, the perturbations in gait, as notable around 0.5 s, could be explained by the changes in the muscle activity pattern (as in the other three steps).

## 4. Discussion

In this paper, we presented a systematic evaluation of the novel IMU-based knee angle measurement device intended for tracking knee activity of children with CP over prolonged periods of time. While most of the state-of-the-art systems use native IMU outputs comprising the linear acceleration, angular velocity, and magnetic field [30,31,32,33,34,35,36,37,38,39], the KAM device uses quaternions to estimate knee angle. The internally calculated and stabilized quaternions were used to estimate the knee angle as the total difference in orientation between two sensors, which is a computationally light operation comprising only basic algebraic operations. Additionally, using quaternions reduces the possibility of reaching Gimbal lock [40]. On the other hand, this makes the system reliant on the internal algorithm (in the IMU chip) which is not fully open, and therefore any potential errors generated within the quaternion calculation and stabilization would propagate to the estimation of the knee angle. Therefore, even after the set of tests shown within this paper, we are not aware of all the possible scenarios that might result in higher errors in the angle estimation.

As shown in the very controllable test with an industrial robot, the KAM device angle estimation is highly linear (R-squared~1) and accurate (a full range error smaller than 1°). Furthermore, even with the misaligned sensors, the device maintained high linearity and accuracy, ensuring that the KAM device will be applicable in real-world use where the placement of the sensors is carried out less rigorously than in a research study. The angle error estimated in the test with the robot showed more resilience to the misalignment between sensors than the error reported in a previous study that focused on the sensor-to-segment misalignment using a set of quick movements [30].

The statistical analysis provided evidence that the KAM device’s knee angle measurements were highly correlated with the optical motion-tracking system, with median cross-correlation coefficients exceeding 0.95 for all walking/running speeds. Such a high cross-correlation is not surprising, as similar results have been reported in previous studies where IMUs were used to estimate joint angles, such as in Bakhshi et al. [32] where the cross-correlation was between 0.94 and 0.99 for different movement tasks, and in Dorschky et al. [37] where the cross-correlation was 0.99 during both walking and running tasks. The mean error that was observed in our study was below 4°, while in the extreme cases, it went to 8°. These results are slightly above the mean absolute differences reported by Schulze et al. [35], where such errors were between 2° and 6.3° for walking speeds between 0.28 m/s and 0.83 m/s, and by Bakhshi et al. [32], where the error was between 0.08° and 2.4° for different movement tasks.

The RMSE was used as the primary evaluation metric in most of the papers presenting similar measurement systems, and therefore, an extensive comparison with the state-of-the-art devices is possible through this metric. In our study, the median RMSE for all subjects was between 5.5° and 8° depending on the walking/running speed, see Figure 7. The study carried out by Seel et al. [33] reported an average RMSE between the IMU-based and optical systems in the range of 0.71° to 3.3°. Similarly, in the study by Yi et al. [41], the RMSE was around 2.5° for the knee joint, while in the study by Fan et al. [30], the RMSE during relatively short and quick movements was around 1°. Furthermore, several studies reported higher RMSEs in the tasks that focus on different activities of daily living. In the study by Niswander et al. [42], the RMSE was 6.35° during the timed-up-and-go task. The study by Tadano et al. [36] reported an average RMSE of 7.88° (the maximum going to 10.7°). The RMSE in the study by Dorschky et al. [37] was around 5.5° during walking and running tasks. Therefore, the comparison with the state-of-the-art knee angle measurement systems based on IMU sensors shows that the KAM device fits within the errors of similar devices during various movement tasks.

Apart from the estimation of the knee angle, the KAM device can measure muscle activity, therefore completing the assessment related to the movement of the knee joint. In this way, it is possible to extract additional information regarding the specific mobility issues, which is especially valuable in the case of persons diagnosed with CP. Dynamic task such as walking and running can be analyzed with the KAM device. Furthermore, it can also be used to analyze sedentary periods during which the knee is fixed in a certain position, such as sitting or sleeping. This feature of KAM goes beyond state-of-the-art devices and presents a major novelty. As presented in Figure 8, the EMG electrodes placed on leg muscles were able to obtain high-quality signals from the knee and ankle muscles. The median signal-to-noise ratio of 8 EMG channels, which was 108 dB in the study presented here, is well beyond the common EMG recording recommendations and practices [43]. Except for excellent amplitude discrimination, the recorded EMG exhibited distinguishable and synchronized activity patterns corresponding to gait phases, indicating accurate temporal muscle activation detection. Within the scope of this paper, temporal muscle activations during self-paced walking were not analyzed, and hence, muscle synergies and other possibly clinically useful EMG features were not systematically evaluated.

The design of the KAM device was made with the intention of providing 24 h continuous monitoring of knee joint motion and muscle activity. For that purpose, an SD card with 32 GB storage is an integral part of the system. As for the power supply, the battery required for continuous 24 h operation is around 5000 mAh for a 3.7 V voltage supply. Having such a relatively large battery cell together with the printed circuit board and onboard chips would significantly increase the size and weight of the KAM device, which will be located in a leg garment. Consequently, it would reduce comfort and possibly impede the natural joint movement. Therefore, it was decided to use an external power bank placed in a pocket and connected to the KAM device via a USB cable. Except for reducing weight on the knee, this solution also enables higher capacity batteries to be used, easier charging, and the possibility of swapping the batteries during operation with minimal loss of data.

One of the limitations of the study is that all the tests with human subjects were carried out on adult persons with no gait-related issues, while the main aim of the device is to measure the activity of the knee joint of children with CP. Although the footprint and weight of the device are reasonably minimal, it still has to be evaluated if the current level of component integration is sufficient to be worn by children without impeding their movement. Furthermore, a specific garment should be designed and manufactured that will hold the KAM device, especially the IMU sensors, at a suitable position on the leg. In our future study, we will focus primarily on adjusting/minimizing the hardware and producing a comfortable garment that will hold the KAM device and the EMG electrodes, preferably with the EMG electrodes already integrated into the garment fabric, analogous to the device presented by Liu et al. [44].

Although this study presents a novelty in the context of evaluating single-joint dynamics through the synchronous measurement of the joint angle and muscle activity, similar concepts were utilized for different purposes. For example, Liu et al. [45,46] constructed a wearable knee bandage system comprising accelerometers, gyroscopes, EMG amplifiers, a goniometer, and a microphone, which proved to be an effective solution for real-time detection of human activities. Conversely, the KAM device was initially created with the primary objective of measuring knee activity in individuals with CP; however, its versatility goes beyond this specific use-case scenario. This versatility means that the device can be applied in a broader context, making it suitable for various applications that require the evaluation of human joints. Whether it is monitoring joint function, detecting activities of daily living, assessing movement disorders, or tracking the progress of rehabilitation, this device can offer a solution that can benefit a wide range of individuals and healthcare professionals.

## Figures and Tables

**Figure 1 sensors-23-08955-f001:**
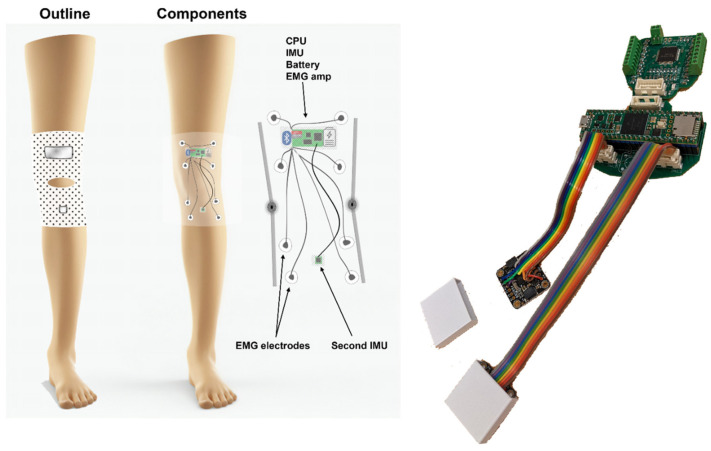
(**Left**) Device overview. The goal of the KAM system is to continuously measure the knee angle over prolonged periods (e.g., 24 h) using light and non-obstructive sensors. Therefore, two miniature IMUs were placed above and below the knee. In addition, the muscle activity of the knee and ankle muscles was measured by an onboard EMG amplifier/digitalizer module. (**Right**) The KAM device hardware. The modularity of the system is provided by the ability of the PCB to be broken into several pieces, one for the microcontroller and connectors, one for the EMG amplifier, and two belonging to IMU boards. This way it is possible to integrate it into the garment in an optimal way, or for example, by removing the EMG part if it is not necessary for a specific study/patient.

**Figure 2 sensors-23-08955-f002:**
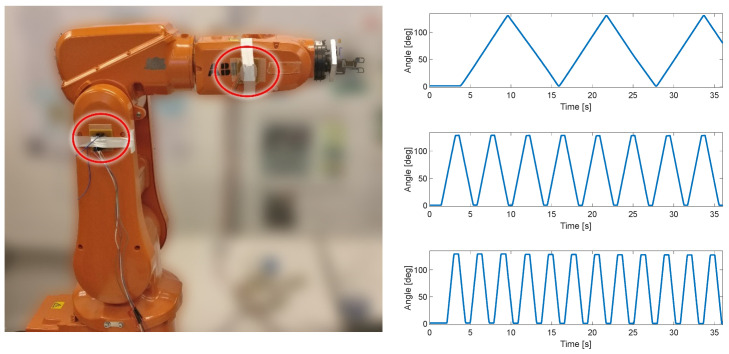
Test setup with ABB IRB120 robot. Two IMU sensors were placed in two segments of the robot (circled in red). In the repeated test, the IMU on the right was rotated by 10° by the actuator to simulate misalignment between two sensors. Three linear angular velocities were produced by the robot, 22°/s, 75°/s, and 150°/s.

**Figure 3 sensors-23-08955-f003:**
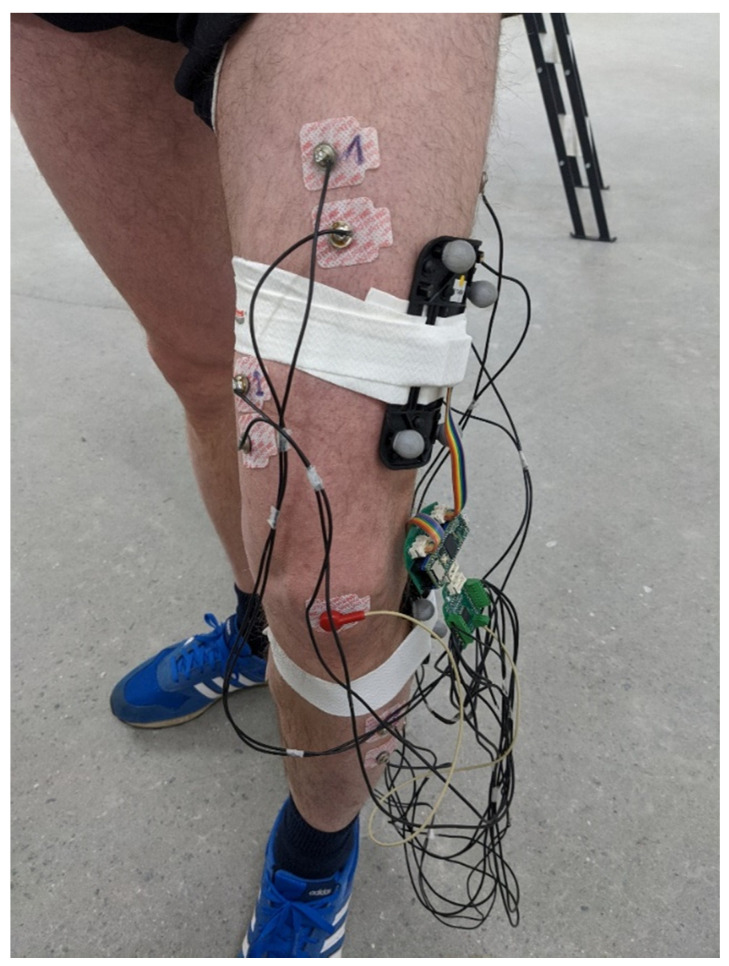
Placement of the EMG electrodes. The 3D-printed templates comprising 5 markers for the motion-tracking system and IMU were placed on the thigh and shank. The KAM device was placed inside a custom-made 3D-printed enclosure and was fixed to the thigh during the measurements using adhesive tape. Four of the EMG channels were picking up signals for thigh muscles, while the remaining four channels were placed on the muscles in the shank.

**Figure 4 sensors-23-08955-f004:**
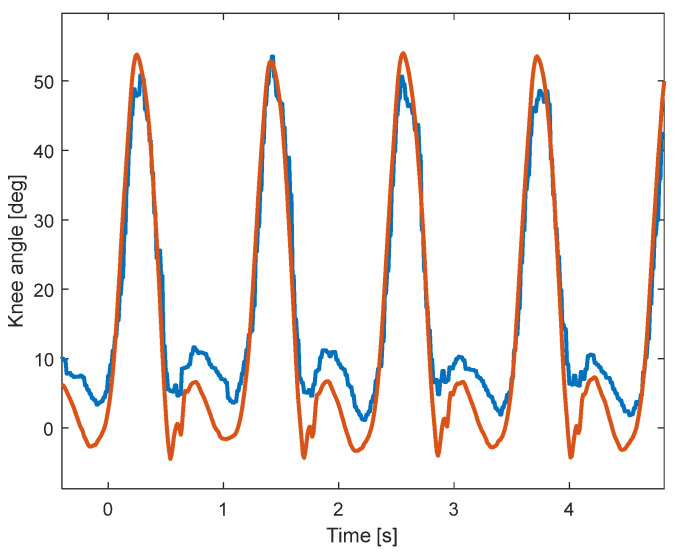
Sample signals during walking at 4 km/h. The knee angle measured by the motion-tracking system (red) is smoother and has a larger range of motion than the knee angle measured by the KAM system (blue).

**Figure 5 sensors-23-08955-f005:**
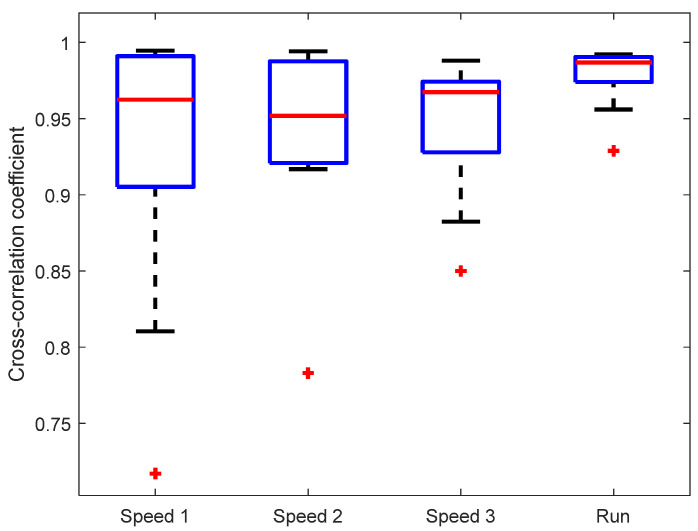
Cross-correlation between knee angles measured by KAM and the motion-tracking system. Marker (+) denotes outlier values.

**Figure 6 sensors-23-08955-f006:**
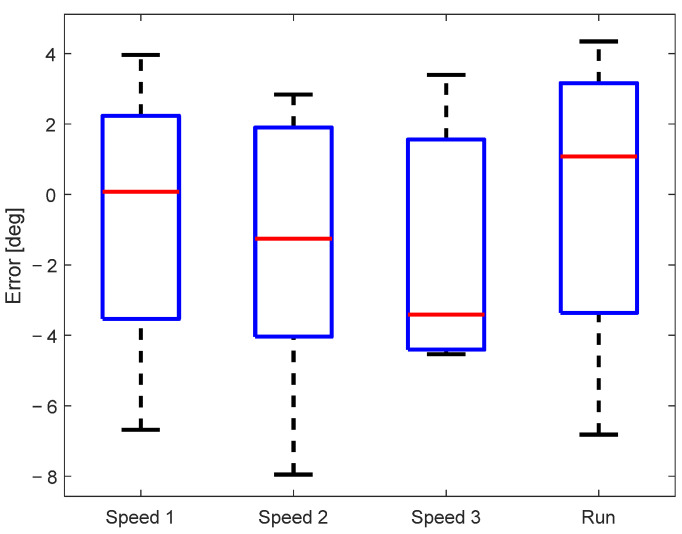
Sample-to-sample error between signals measured by the motion-tracking and KAM systems.

**Figure 7 sensors-23-08955-f007:**
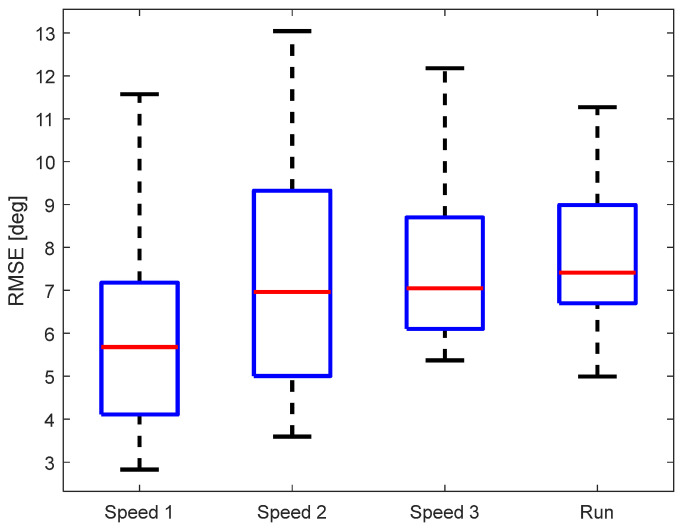
RMSE for the different walking/running speeds.

**Figure 8 sensors-23-08955-f008:**
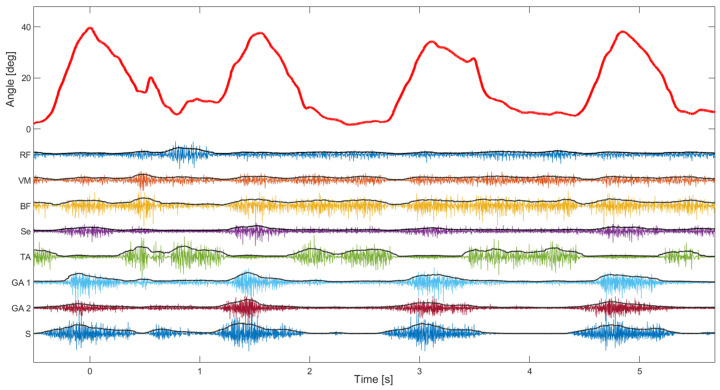
Knee angle and 8 channels of EMG during slow walking. The angle (red signal) was shown during four consecutive steps. On the same plot are superimposed EMG channels and their envelopes (RF—Rectus femoris, VM—Vastus medialis, BF—Biceps femoris, Se—Semitendinosus, TA—Tibialis anterior, GA—Gastrocnemius, and S—Soleus).

**Table 1 sensors-23-08955-t001:** Results of the tests using the industrial robot. Slow, medium, and fast denote angular velocities of 22°/s, 75°/s, and 150°/s, respectively. Tilt denotes test conditions when one of the IMUs was misaligned/rotated with respect to the other by 10°.

	FRE [°]	SSE	R-Squared	RMSE
Slow	0.44	8.9	0.99	0.15
Medium	0.92	2.1	0.99	0.11
Fast	0.7	1.2	0.99	0.12
Slow-Tilt	0.25	5.5	0.99	0.09
Medium-Tilt	0.81	0.5	0.99	0.05
Fast-Tilt	0.41	1.1	0.99	0.12

## Data Availability

The data presented in this study are available on request from the corresponding author. The data are not publicly available due to ethical restrictions.

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
