# Peer review of "An Integrated Approach for Real-Time Monitoring of Knee Dynamics with IMUs and Multichannel EMG"

_sensors, 2023, doi:10.3390/s23218955_

Round 1

Reviewer 1 Report

Comments and Suggestions for Authors

An interesting work. It focuses on health and the well-being of our children. I appreciate it. I do not have doubts on hardware, data, models, and results, but rather some structural concerns, which is, however, also critical:

The title and abstract/introduction sections of the manuscript are disconnected. The title does not emphasize the problem and motivation (CP) of the study, but the article as a whole begins with the formulation of the CP problem as the sole aim. One of the following two points needs to be chosen:

1. the title of the current manuscript is confusing and wastes clicks from readers who are only interested in basic IMU-EMG-based knee research. The title should be corrected and narrowed to align it with your focus of attention (CP).

2. or the authors should emphasize the significance and relevance of their work to the broader IMU-EMG-based knee research in the introduction (and abstract) section, rather than just limiting it to CP, so that the current title can be used without academic error.

I would prefer the second. It could have made your work more accessible to a wider audience. The application of IMU and EMG to research on the knee is a widely valued endeavor, not necessary to be linked only to CP. Some review of work in related fields could be added to make the title more reasonable and device application scenarios more valuable. This and last year's SOTA included the integration of multiple IMUs and EMGs on knee bandages, as well as various other types of sensors (e.g., goniometers for measuring angles), to conduct HAR studies (On a Real Real-Time Wearable Human Activity Recognition System) for disease diagnosis, rehabilitation assistance, and motion duration analysis (How Long Are Various Types of Daily Activities). Besides their own medical meanings, these fields are also potentially relevant for CP. For example, the duration of daily activities obtained through IMU-EMG-based HAR may be used as a novel reference for CP analysis and treatment.

Incidentally, the scheme of placing two IMUs and four EMGs in Biosignal Processing and Activity Modeling for Multimodal Human Activity Recognition (e.g., Fig 1.3 and Tab 3.2) has a high degree of similarity to your schematic; however, it uses a knee bandage as the carrier for the wearability for future productization. Although it has different goals, it is a highly informative paradigm for future IMU-EMG-based CP-related products (e.g., the future garment design you mentioned). This type of work should not be ignored from the perspective of sensor placement and integration.

Please distinguish between the parameter name and the textual part (\text) in the equations. For example, RMS, signal, and noise should be text content.

Author Response

A: We would like to thank the reviewer for providing very important and constructive comments that helped us improve the quality of the manuscript. Our point-to-point replies are provided below.

R: An interesting work. It focuses on health and the well-being of our children. I appreciate it. I do not have doubts on hardware, data, models, and results, but rather some structural concerns, which is, however, also critical:

The title and abstract/introduction sections of the manuscript are disconnected. The title does not emphasize the problem and motivation (CP) of the study, but the article as a whole begins with the formulation of the CP problem as the sole aim. One of the following two points needs to be chosen:

  1. the title of the current manuscript is confusing and wastes clicks from readers who are only interested in basic IMU-EMG-based knee research. The title should be corrected and narrowed to align it with your focus of attention (CP).
  2. or the authors should emphasize the significance and relevance of their work to the broader IMU-EMG-based knee research in the introduction (and abstract) section, rather than just limiting it to CP, so that the current title can be used without academic error.

I would prefer the second. It could have made your work more accessible to a wider audience. The application of IMU and EMG to research on the knee is a widely valued endeavor, not necessary to be linked only to CP. Some review of work in related fields could be added to make the title more reasonable and device application scenarios more valuable. This and last year's SOTA included the integration of multiple IMUs and EMGs on knee bandages, as well as various other types of sensors (e.g., goniometers for measuring angles), to conduct HAR studies (On a Real Real-Time Wearable Human Activity Recognition System) for disease diagnosis, rehabilitation assistance, and motion duration analysis (How Long Are Various Types of Daily Activities). Besides their own medical meanings, these fields are also potentially relevant for CP. For example, the duration of daily activities obtained through IMU-EMG-based HAR may be used as a novel reference for CP analysis and treatment.

A: Thank you for a very constructive and insightful comment. We agree that the second point is more suitable for the device we developed and tested within this study. Therefore in the revised manuscript, we provided a broader background and extended the potential application of the device to other use cases. Hence, we added text in the Abstract, Introduction, and Discussion sections.

R: Incidentally, the scheme of placing two IMUs and four EMGs in Biosignal Processing and Activity Modeling for Multimodal Human Activity Recognition (e.g., Fig 1.3 and Tab 3.2) has a high degree of similarity to your schematic; however, it uses a knee bandage as the carrier for the wearability for future productization. Although it has different goals, it is a highly informative paradigm for future IMU-EMG-based CP-related products (e.g., the future garment design you mentioned). This type of work should not be ignored from the perspective of sensor placement and integration.

A: Thank you for providing a very valuable comment as we were not aware of this device (due to our focus on CP and knee angle measurement). This is indeed a very relevant study which we acknowledged in the revised manuscript.

 Added text:

“In our future study, we will focus primarily on adjusting/minimizing the hardware and producing a comfortable garment that will hold the KAM device and the EMG electrodes, preferably with the EMG electrodes already integrated into the garment fabric, analogous to the device presented by Liu et al. [44].”

and

“Although this study presents a novelty in the context of evaluating a single joint dynamics through the synchronous measurement of joint angle and muscle activity, similar concepts were utilized for different purposes. For example, Liu et al. [45,46] constructed a wearable knee bandage system comprising accelerometers, gyroscopes, EMG amplifiers, a goniometer, and a microphone, which proved to be an effective solution for real-time detection of human activities.”

R: Please distinguish between the parameter name and the textual part (\text) in the equations. For example, RMS, signal, and noise should be text content.

A: Thank you for pointing out this error. We corrected the equations.

Reviewer 2 Report

Comments and Suggestions for Authors

In this work, the authors focused on addressing the need for accurate measurement of knee angles in individuals with cerebral palsy (CP) during gait and daily activities. They introduced a novel multimodal approach that combines Inertial Measurement Unit (IMU) sensors and Electromyography (EMG) for this purpose. The study evaluated the performance of this integrated approach, emphasizing the accuracy of IMU sensors in capturing knee joint movements compared to an optical motion tracking system. They also highlighted the additional insights provided by EMG in assessing muscle activation patterns. Although the research aimed to improve the accuracy of knee angle measurement in individuals with CP, no experiments have been shown with CP individuals. In addition, there are several technical and organizational concerns that need to be addressed.

1. The authors have targeted their work, from abstract to discussions to show the benefits of the proposed integrated approach for CP individuals, however, they tested the approach with able-bodied individuals. Either the authors are suggested to provide the results for CP or revise the language everywhere carefully.

2. In the Introduction section, the authors are suggested to highlight the limitations of the existing motion capture devices and related approaches, and then tell clearly what are their novel contributions. 

3. It seems from the methodology section that the authors have used IMUs with EMG sensors to estimate muscle activity in addition to the knee joint angles. However, it is completely unclear why there is a need to integrate them in such a way. One can measure the joint angles in parallel with the EMG signals. The advantages of using an integrated approach are missing and misleading at the same time.

4. In lines 168-173, the authors have tested the IMUs only with the robotic arm. Then, they can not write they tested the knee angle measurement (KAM) device with the robotic arm since they have not used the EMG sensors with the robotic arm (which is also obviously not possible). Please revise the text carefully. 

5. The authors are suggested to include inclusion and exclusion criteria for the healthy subjects. Moreover, there should be enough information on the experiment protocol such as no. of speeds (you have mentioned different walking/running speeds).

6. The explanation of results is limited. The authors should explain properly the insights from Figures 5-7. What is interesting in Figure 8? It is obvious that muscle activity will be evident when changing the gait phases. How does using the EMG sensors complement angle measurement results?

7. The authors are strongly recommended to provide a quantitative comparison of the results from existing works in the literature. At present, it is hard to interpret the novelty of the proposed work.

Author Response

A: We would like to thank the reviewer for providing very important and constructive comments that helped us improve the quality of the manuscript. Our point-to-point replies are provided below.

R: 

In this work, the authors focused on addressing the need for accurate measurement of knee angles in individuals with cerebral palsy (CP) during gait and daily activities. They introduced a novel multimodal approach that combines Inertial Measurement Unit (IMU) sensors and Electromyography (EMG) for this purpose. The study evaluated the performance of this integrated approach, emphasizing the accuracy of IMU sensors in capturing knee joint movements compared to an optical motion tracking system. They also highlighted the additional insights provided by EMG in assessing muscle activation patterns. Although the research aimed to improve the accuracy of knee angle measurement in individuals with CP, no experiments have been shown with CP individuals. In addition, there are several technical and organizational concerns that need to be addressed.

  1. The authors have targeted their work, from abstract to discussions to show the benefits of the proposed integrated approach for CP individuals, however, they tested the approach with able-bodied individuals. Either the authors are suggested to provide the results for CP or revise the language everywhere carefully.

A: Thank you for the constructive comment. We agree that the paper was too focused on CP while presenting only results from able-bodied individuals. Therefore, in the revised manuscript we have broadened the potential application of the presented device as neither, the hardware, nor the algorithms are made specifically for evaluating CP. We introduced the changes in the Abstract, Introduction, and Discussion sections to reflect on these potential applications.

  1. In the Introduction section, the authors are suggested to highlight the limitations of the existing motion capture devices and related approaches, and then tell clearly what are their novel contributions.

A: Thank you for your very valuable comment. We agree that we have not explicitly provided a knowledge gap within the Introduction section. Therefore, we added a new paragraph providing a shortcoming of the state-of-the-art and the motivation for the development presented in the paper.

Added text:

“As shown in the previous paragraph, both knee angle and muscle activity provide valuable and complementary information about the spasticity and contractures of the knee joint. However, the previous studies focused on one out of these two aspects. To fully understand and evaluate knee use and the progression of CP-induced issues, it is beneficial to simultaneously obtain the EMG signal and the resulting knee angle over prolonged periods of time, such as for 24 hours. That way it is possible not only to detect insufficient stretching of the knee during a day, but also to assess muscle activation patterns in dynamic and sedentary activities of daily living. Importantly, using this multimodal measurement it is possible to know if the muscles are contracted, co-contracted or relaxed in periods when the knee was static at a certain angle, i.e. during sleeping, which was missing from the previous studies.”

  1. It seems from the methodology section that the authors have used IMUs with EMG sensors to estimate muscle activity in addition to the knee joint angles. However, it is completely unclear why there is a need to integrate them in such a way. One can measure the joint angles in parallel with the EMG signals. The advantages of using an integrated approach are missing and misleading at the same time.

A: Thank you for pointing this out. The main aim of the developed device was to fulfill the requirements necessary for conducting a clinical study with patients with CP. Namely, we needed a miniature device that will be used by children with CP, which can measure knee angle and EMG, and log the data for 24 hours. This is the primary reason for making a novel device instead of using two, or three systems that can do the same function. This was stated in section 2.2.

  1. In lines 168-173, the authors have tested the IMUs only with the robotic arm. Then, they can not write they tested the knee angle measurement (KAM) device with the robotic arm since they have not used the EMG sensors with the robotic arm (which is also obviously not possible). Please revise the text carefully.

A: Thank you for the comment. To explicitly state the protocol for this evaluation we added the following sentence in section 2.3:

„This test included only the angle measurement module to estimate the angle between two IMU sensors.“

  1. The authors are suggested to include inclusion and exclusion criteria for the healthy subjects. Moreover, there should be enough information on the experiment protocol such as no. of speeds (you have mentioned different walking/running speeds).

A: Thank you for pointing this out. We added the Exclusion criterion in the text:

„The exclusion criterion was the inability of the participant to maintain the tread-mill-imposed walking speed for 10 minutes.“

As for the walking/running speeds, the information related to them is provided in section 2.3, Lines 225-226 in the revised manuscript.

  1. The explanation of results is limited. The authors should explain properly the insights from Figures 5-7. What is interesting in Figure 8? It is obvious that muscle activity will be evident when changing the gait phases. How does using the EMG sensors complement angle measurement results?

A: Thank you for identifying this shortcoming. We made additional comments on Figures 5-8. We hope that the added text provides relevant information to support the figures.

  1. The authors are strongly recommended to provide a quantitative comparison of the results from existing works in the literature. At present, it is hard to interpret the novelty of the proposed work.

A: In the original submission we dedicated two paragraphs in the Discussion section where we extensively compared all metrics from our paper to the state-of-the-art. In our opinion, we provided a fair and broad comparison. Nevertheless, if we omitted some specific quantitative metric related to an aspect of our device, we would gladly introduce and discuss it in the next revision of the manuscript. 

Round 2

Reviewer 1 Report

Comments and Suggestions for Authors

The authors addressed my concerns and I argue to accept the manuscript.

Reviewer 2 Report

Comments and Suggestions for Authors

The authors have addressed all the concerns raised by the reviewer. The manuscript could be accepted without any further revisions.